# Role of SOCS and VHL Proteins in Neuronal Differentiation and Development

**DOI:** 10.3390/ijms24043880

**Published:** 2023-02-15

**Authors:** Hiroshi Kanno, Shutaro Matsumoto, Tetsuya Yoshizumi, Kimihiro Nakahara, Atsuhiko Kubo, Hidetoshi Murata, Taro Shuin, Hoi-Sang U

**Affiliations:** 1Department of Neurosurgery, School of Medicine, Yokohama City University, Yokohama 232-0024, Japan; 2Department of Neurosurgery, Asahi Hospital, Tokyo 121-0078, Japan; 3Department of Neurosurgery, St. Mariannna Medical University, Kawasaki 216-8511, Japan; 4Department of Neurosurgery, International University of Health and Welfare, Atami 413-0012, Japan; 5Nerve Care Clinic, Yokosuka 238-0012, Japan; 6Kochi Medical School Hospital, Nangoku 783-0043, Japan; 7Department of Electrical Engineering, University of California San Diego, San Diego, CA 92093, USA

**Keywords:** development, JAK-STAT pathway, neuronal differentiation, SOCS protein, ubiquitination, VHL protein

## Abstract

The basic helix–loop–helix factors play a central role in neuronal differentiation and nervous system development, which involve the Notch and signal transducer and activator of transcription (STAT)/small mother against decapentaplegic signaling pathways. Neural stem cells differentiate into three nervous system lineages, and the suppressor of cytokine signaling (SOCS) and von Hippel-Lindau (VHL) proteins are involved in this neuronal differentiation. The SOCS and VHL proteins both contain homologous structures comprising the BC-box motif. SOCSs recruit Elongin C, Elongin B, Cullin5(Cul5), and Rbx2, whereas VHL recruits Elongin C, Elongin B, Cul2, and Rbx1. SOCSs form SBC-Cul5/E3 complexes, and VHL forms a VBC-Cul2/E3 complex. These complexes degrade the target protein and suppress its downstream transduction pathway by acting as E3 ligases via the ubiquitin–proteasome system. The Janus kinase (JAK) is the main target protein of the E3 ligase SBC-Cul5, whereas hypoxia-inducible factor is the primary target protein of the E3 ligase VBC-Cul2; nonetheless, VBC-Cul2 also targets the JAK. SOCSs not only act on the ubiquitin–proteasome system but also act directly on JAKs to suppress the Janus kinase–signal transduction and activator of transcription (JAK-STAT) pathway. Both SOCS and VHL are expressed in the nervous system, predominantly in brain neurons in the embryonic stage. Both SOCS and VHL induce neuronal differentiation. SOCS is involved in differentiation into neurons, whereas VHL is involved in differentiation into neurons and oligodendrocytes; both proteins promote neurite outgrowth. It has also been suggested that the inactivation of these proteins may lead to the development of nervous system malignancies and that these proteins may function as tumor suppressors. The mechanism of action of SOCS and VHL involved in neuronal differentiation and nervous system development is thought to be mediated through the inhibition of downstream signaling pathways, JAK-STAT, and hypoxia-inducible factor–vascular endothelial growth factor pathways. In addition, because SOCS and VHL promote nerve regeneration, they are expected to be applied in neuronal regenerative medicine for traumatic brain injury and stroke.

## 1. Introduction

Since Cajal, a prominent 19th-century Spanish neuroanatomist, proposed that damaged central nerves cannot regenerate, it has long been believed that stem cells, the source of tissue regeneration, do not exist in the adult central nervous system. However, in 1965, newly generated neurons were observed in the hippocampus of adult rats [1]. Furthermore, in 1998, it was demonstrated that neural stem cells (NSCs) exist in the human subventricular zone (SVZ) and sub-granular zones of the hippocampal dentate gyrus [2], and it became clear that neurons are also produced in the adult human hippocampus. 

To date, the understanding of the factors affecting neural differentiation and the development of the nervous system has developed rapidly via research on the differentiation mechanism of NSCs. Previous studies have identified a large number of cellular extrinsic factors and transcription factors that control NSC differentiation; this understanding facilitates our ability to induce the differentiation of NSCs into target cell types in vitro. For example, differentiation into neurons can be induced by the transcription of the pro-neural gene, neurogenin1 (*Ngn1*), followed by the induction of NeuroD gene expression [3]. The differentiation of NSCs into neurons is induced by promoting gene expression, and basic helix–loop–helix (bHLH)-type transcription factors are important for the differentiation of NSCs into oligodendrocytes and neurons [3,4]. The bHLH-type transcription factors, Olig1 and Olig2, whose expression is induced by Sonic Hedgehog (Shh) signaling, have been shown to regulate differentiation into oligodendrocytes and induce oligodendrocyte maturation [4]. bHLH-type transcription factors are inhibited by HES and ID family molecules [5,6]; furthermore, Notch and bone morphogenetic protein (BMP) signals induce the expression of these inhibitory bHLH factors [7,8], thereby promoting the neuronal differentiation of NSCs and oligodendrogenesis. It is involved in maintaining the undifferentiated state of NSCs and suppressing cell differentiation [7,8]. In contrast, interleukin-6 (IL-6) family cytokines (IL-6FCs) are known to induce the differentiation of neural connective cells into astrocytes [9]. The simultaneous stimulation of NSCs with IL-6FCs and BMP has been shown to synergistically induce differentiation into astrocytes [9]. Crosstalk in signaling pathways has also been revealed. In addition, IL-6FCs induce suppressor of cytokine signaling (SOCS) proteins and inhibit the Janus kinase–signal transduction and activator of transcription (JAK-STAT) pathway [10], thereby inhibiting the differentiation of NSCs into astrocytes. Overall, SOCS and von Hippel-Lindau (VHL) proteins work to suppress the JAK-STAT and/or hypoxia-inducible factor–vascular endothelial growth factor (HIF-VEGF) pathways [11,12], which are downstream of each protein, respectively. They then suppress the differentiation of NSCs into astrocytes and promote their differentiation into neurons and oligodendrocytes (Figure 1). 

In this study, we describe the role of SOCS and VHL proteins in neuronal differentiation and development.

## 2. Structure of SOCS and VHL Proteins

SOCS (CIS and SOCS 1–7) and VHL form the Elongin B/C–Cul2/Cul5–SOCS-box protein (ECS) complex, a member of the ubiquitin ligase family that shares a Cullin-Rbx module. SOCS and VHL bind Cullin-Rbx via BC-box-binding Elongin C; however, VHL lacks the C-terminal sequence of the SOCS box (downstream of the BC box). In mammalian cells, SOCSs bind to Cul5-Rbx2; in contrast, VHL specifically interacts with endogenous Cul2-Rbx1. Therefore, the ECS complex has two distinct protein assemblies: one containing a subunit with a VHL box (composed of a BC box and a downstream Cul2 box) that interacts with Cul2–Rbx1 and the other containing a subunit with a SOCS box (composed of a BC box and a downstream Cul5 box). Domain exchange analyses demonstrated that the specificity of the interaction of VHL-box and SOCS-box proteins with the Cullin-Rbx module was conferred by Cul2 and Cul5 boxes, respectively. Additionally, the RNAi-mediated knockdown of Cul2–Rbx1 inhibited the VHL-mediated degradation of HIF-2α, whereas Cul5–Rbx2 knockdown did not affect the function of the Cul2–Rbx1 and Cul5–Rbx2 modules [13] (Figure 2). 

## 3. Characteristics of SOCS Proteins and Their Roles in Neuronal Differentiation and Nervous System Development

### 3.1. Features and Functions of SOCS Proteins

The discovery of the SOCS family began in 1995 with the isolation of a novel cytokine-inducible Src homology 2 (SH2)-containing protein (CIS) as an early gene induced by IL-2, IL-3, and erythropoietin (EPO) [13]. Subsequently, an SH2 domain–containing protein that inhibits IL-6-induced macrophage differentiation was identified in 1997 and named SOCS1 [14,15,16]. Furthermore, database searches of similar amino acid sequences predicted from SOCS1 revealed that at least 20 proteins from mice and humans share sequence homology within a 40-residue C-terminal motif called the SOCS box. Proteins containing SOCS boxes were grouped into subfamilies based on the domains contained in the central region, and family member proteins containing central SH2 domains were named SOCSs (CIS and SOCS1–7). Each SOCS has three distinct domains: a least-conserved N-terminal domain, a conserved central SH2 domain, and a highly conserved C-terminal SOCS-box domain. The length of the N-terminal domain varies among the members; SOCS1–3 and CIS contain shorter N-terminal domains than in SOCS4–7 [17]. The SOCS-box motif recruits an E3 ubiquitin ligase complex consisting of Elongin B/C, Rbx2, and Cullin5. This complex forms an E3 ubiquitin ligase that ubiquitin-tags target proteins, such as JAKs and cytokine receptors, and degrades them in the proteasome [18,19]. SOCSs regulate signaling by linking substrates to the ubiquitination machinery through the SOCS box. SOCS expression, induced by cytokine stimulation, not only interferes with cytokine signaling in a negative feedback loop but also interferes with other cytokines downstream in a response known as “crosstalk” [20,21]. Currently, various molecules, including numerous cytokines, such as interferons (IFNs), growth factors, and hormone factors, are known to activate the JAK-STAT pathway; in contrast, the JAK-STAT pathway is inhibited by SOCS [22] (Figure 3).

Among the SOCS family members, CIS, SOCS1, and SOCS3 inhibit the JAK-STAT signaling pathway; however, their modes of action are slightly different. SOCS1 directly interacts with JAK1, JAK2, and tyrosine kinase 2 (Tyk2) and inhibits their phosphorylation and catalytic activity, thereby downregulating the JAK-dependent phosphorylation of receptors and the signal transducer and activator of transcription (STATs) [19,20,23].

Structurally, the SH2 domain and N-terminal region of SOCS1 are essential for inhibition of JAK activity, whereas the C-terminal SOCS-box has been shown to be dispensable [23,24,25].

CIS expression is induced by EPO [24], growth hormone (GH) [25,26,27,28,29], IL-2 [30,31,32,33,34], IL-3 [31,32], and prolactin (PRL) [32,33]. SOCS1 expression also inhibits IFN signaling by interacting with the IFN-α/β receptor subunit 1 and IFN-γ receptor subunits, which in turn inhibit IFN activation via STAT3 [34]. Additionally, SOCS1 promotes the degradation of p65 in the NF-κB signaling pathway and reduces the kinase apoptotic signal that regulates kinase 1 upstream in the JNK and p38 signaling pathways; consequently, SOCS1-mediated degradation inhibits GH [35,36,37], IFN [29,30,31], IL-4 [32,33,34], IL-6 [29,32], thrombopoietin (TPO) [29], PRL [38]. SOCS2 expression is then also induced by STAT5-activating cytokines such as ciliary neurotrophic factor (CNTF) [39], IFNα [40], IFNγ [18,41], leukemia inhibitory factor (LIF) [18], IL-1β [41], IL-4 [18], IL-6 [41], IL-15 [42], and insulin (INS) [43]. In contrast, SOCS2 downregulates the expression of the cytokines GH [44], PRL [45], LIF [45], IL-2, IL-3 [46], IL-6 [47], epidermal growth factor (EGF) [48], and insulin-like growth factor (IGF)-1 via a negative feedback loop [49].

SOCS3 has been demonstrated to be induced by the cytokines IL-1β [50], IL-2 [51], IL-3 [52], IL-4 [32], IL-6 [53], IL-9 [54], IL-10 [55], IL-11 [56], IL-13 [18], IL-22 [57], IFN-γ [31], IFNα [31], EPO [38], LIF [58], PRL [34], GH [58], leptin [59], G-CSF [18], GM-CSF [60], CNTF [39], TPO [61], TNFα [62], CT-1 [63], and oncostatin M signaling (OSM) [64]. It is also induced by several growth factors, including EGF [65], platelet-derived growth factor (PDGF) [65], thyroid-stimulating hormone (TSH) [66], insulin [43], and basic fibroblast growth factor (bFGF) [67]. SOCS3 has been demonstrated to play a regulatory role in the downstream signaling of a wide range of cytokine receptors, including those for IL-2 [42], IL-6 [47], IL-9 [54], IL-11 [56], IL12 [68], IL-23 [69], IL-27 [70], IFNα/β [31], IFNγ [31], G-CSF [71], EPO [34], PRL [34], GH [72], LIF [58], leptin [59], CNTF [61], IL-1β [73], OSM [65], and CT-1 [64], as well as IGF-1 [74], INS [75], CD28 [27] and calcineurin [76]. Conversely, SOCS3 inhibits the nuclear factor κB (NF-κB) signaling pathway, antagonizes cyclic AMP (cAMP)-mediated signaling, and promotes signaling via the microtubule affinity-regulating kinase (MAPK) pathway [77]. 

SOCS4 has been shown to be induced exclusively by EGF [78] and has been demonstrated to regulate EGFR signaling in vitro [79]. This induction is mediated by SOCS4 docking on phosphotyrosine residues on activated EGFR and the subsequent targeting of the receptor for proteasomal degradation by the recruitment of E3 ubiquitin ligase activity [79,80]. Similar to STAT3, SOCS4 binds to EGFR phosphotyrosine with a high affinity. In addition, SOCS4 regulates primordial follicle activation initiated by LIF activation of the JAK1-STAT3 pathway and interacts with several proteins involved in follicle development [81]. 

SOCS5 expression has been shown to be induced by EGF in vitro [78]. SOCS5 regulates both receptor tyrosine kinase (RTK) and cytokine receptor signaling and negatively regulates EGFR in vitro [80,81]. In addition, SOCS5 downregulates the receptors of IL-6, LIF [82], and IL-4 signaling [78,79]. SOCS5 has also been hypothesized to regulate signaling by initiating IL-4 signaling. SOCS5 has also been implicated in T helper (Th) cell differentiation, particularly in the balance between Th1 and Th2 cells. SOCS5 inhibits activation of STAT6 by the IL-4 receptor and typically induces T-cell differentiation toward a Th2 fate [83,84]. Additionally, SOCS5 transgenic mice show increased IL-2 and IFN-γ expression [85]. SOCS6 is induced by INS [85,86], IGF-2 [87], IGF-1 [87], FMS-like tyrosine kinase 3 (FLT3) [88], and stem cell factor (SCF) [89]. SOCS5 also negatively regulates T-cell antigen receptor (TCR) signaling [90]. 

SOCS6 exerts its regulatory effects predominantly through the ubiquitination and degradation of target proteins [41]. SOCS6 also interacts with another E3 ligase component, heme-oxidized IRP2 ubiquitin ligase-1 (HOIL-1) [89]. Further, the SOCS6 N-terminal domain promotes localization to the nucleus, where it negatively regulates STAT3. The overexpression of SOCS6 [90] results in the repression of TCR-dependent IL-2 promotion and activation of MAPK pathway factors, such as ERK1/2 and p38 [34]. SOCS6 also binds to FLT3 and negatively regulates its corresponding signaling [88]. Furthermore, SOCS6 has been shown to inhibit the downstream pathways of INS and IGF-1 receptors [86]. Specifically, SOCS6 interacts with PIM3, a protein upregulated in β-cells in response to glucose stimulation. 

SOCS7 is induced by the cytokines GH, PRL [41], EGF [91], INS, and IGF-1 [41] and regulates signaling via GH, PRL, leptin [91], and INS [92]. SOCS7 also inhibits the nuclear transport of the following adapter proteins: non-catalytic region of tyrosine kinase (NCK) [93], insulin receptor substrate (IRS)-1 [91], IRS-2 [92], IRS-4 [92], p85 phosphatidylinositol 3-kinase (PI3K) [92], and growth factor receptor-bound protein 2 (GRB2) [93]. SOCS7 regulates signaling through the recruitment of E3 ubiquitin ligase activity and subsequent proteasome targeting of associated proteins [92]. SOCS7 interacts with INS receptors and their adapter proteins, playing an active role in insulin signaling [92,93]. In addition, associations have been observed between SOCS7 haplotypes and various metabolic traits, such as obesity, insulin resistance, and lipid metabolism [94]. 

### 3.2. SOCS Gene and Protein Expression in the Nervous System

Members of the SOCS family may play important roles in the nervous system, primarily owing to their regulatory action on the JAK-STAT pathway, which is involved in nervous system development and differentiation. The expression of SOCS1, SOCS2, and SOCS3 in the development of adult nervous systems was investigated using northern analysis and in situ hybridization [95]. In the corresponding northern analysis, all three genes were expressed in the brain, with SOCS2 being the most highly expressed; maximal expression was observed from embryonic day 14 to postnatal day 8 and decreased thereafter. In situ hybridization analysis revealed that the expression patterns of SOCS1 and SOCS3 were low and widespread, whereas SOCS2 was only expressed in neurons; the corresponding expression of SOCS2 was induced during neuronal differentiation [95]. SOCS4 is widely expressed in the brain, with a particularly high expression in the olfactory nerve. SOCS5 is not only highly expressed in the placenta but also in the brain [21,79,96,97]. In contrast, in the nervous system, SOCS6 and SOCS7 are highly expressed in the retina and throughout the brain, respectively [92,93].

### 3.3. Role of SOCS Proteins in the Molecular Mechanisms of Neuronal Differentiation and Nervous System Development

SOCSs are involved in the differentiation of various cells. Regarding T-cell differentiation, SOCS1 in Th cells is induced by IL-6 and promotes Th17 differentiation [98]. In contrast, SOCS3 promotes the differentiation of myeloid cells into lung tissue CD8+ T lymphocytes via Notch1 upregulation [99]. SOCSs also play important roles in neuronal differentiation and regeneration, neurogenesis, and nervous system development. SOCS1 regulates IFN-γ-mediated survival of sensory neurons [100] and NSC proliferation and differentiation into neurons [101,102].

SOCS2 is a negative regulator of GH signaling [103,104,105]. In contrast, SOCS2 was shown to promote neurite outgrowth, regulate neuronal morphology, induce neurogenesis, and inhibit NSC astrogliogenesis [103,106]. SOCS3 associates with IGF receptors and inhibits the JAK-STAT pathway [107]. Furthermore, SOCS3 induces neuronal differentiation and promotes the neuronal survival of PC12 cells [108]. SOCS6 also promotes the neuronal differentiation of NSCs [87]. Additionally, we showed that the BC-box motif in SOCS6 promotes neuronal differentiation into GABAergic neurons [109].

The neurogenic cytokine family, which includes IL-6, LIF, and CNTF, also regulates the JAK-STAT pathway. The neurogenic cytokine family proteins are activated to alter NSC self-renewal and progenitor cell division and differentiation; furthermore, these proteins induce SOCS expression to exert negative feedback on the JAK-STAT pathway. Experiments with NSCs isolated from the adult mouse SVZ showed that both cytokines activated the JAK-STAT pathway and stimulated NSC self-renewal when LIF or CNTF was administered in the culture medium. Thus, SOCS promotes the growth and development of NSC neurospheres. Consequently, LIF treatment for acute brain injury has been observed to promote SVZ regeneration, possibly by increasing the number of NSCs [110].

The classical neurotrophins nerve growth factor (NGF), brain-derived neurotrophic factor, neurotensin (NT)-3, and NT-4 regulate multiple aspects of neuronal differentiation, survival, and growth. To exert these effects, these factors specifically bind to receptors of the tropomyosin-related kinase (Trk) receptor tyrosine kinase family, namely TrkA, TrkB, and TrkC. They are internalized and localized in various cellular compartments where signaling occurs. Conversely, among the SOCS family members, SOCS2, which is also a regulator of NGF signaling, alters the subcellular localization and downstream signaling of TrkA and plays a role in neuronal differentiation and neurite outgrowth, similar to NGF [111].

The expression of SOCS1 and SOCS3 in glioblastoma (GBM) cells is decreased. In addition, the JAK-STAT3 signaling pathway is activated in human gliomas, leading to poor prognosis [112]. SOCS1 and SOCS3 may also function as epigenetic regulators of GBM hypermethylation, associated with poor prognosis in GBM [113]. NF-κB-driven SOCS expression functions as a negative regulator of the JAK-STAT signaling pathway. In GBM, E3 ubiquitin ligases are involved in regulating cellular functions, such as RTK survival signals. SOCSs negatively regulate RTK signaling, and kinase overexpression or mutation is associated with glioma malignancy [114].

## 4. Characteristics of VHL Proteins and Their Roles in Neuronal Differentiation and Nervous System Development

### 4.1. Features and Functions of VHL Proteins

VHL is the product of the VHL tumor suppressor gene, the causative gene of VHL disease. VHL disease is an autosomal dominant hereditary multiple neoplastic disease that causes hemangioblastoma, retinal hemangioma, renal cell carcinoma, pheochromocytoma, and pancreatic, renal, and epididymal cysts. This disease was named after Eugene von Hippel, who reported familial retinal angioma in 1904, and Arvid Lindau, who reported a case of retinal hemangioma and cerebellar hemangioblastoma in 1926 [115].

VHL disease is caused by mutations in the VHL tumor suppressor gene. *VHL* was isolated and identified from the 3p25 region using the positional cloning method described by Zbar et al. [116]. *VHL* consists of 639 base pairs and three exons. The genetic diagnosis of patients with this disease using this gene revealed mutations concentrated in the binding site of the binding protein. In approximately 20% of cases, the entire *VHL* locus is deleted, which can be detected by quantitative Southern blot analysis or fluorescence in situ hybridization [115,116]. Approximately 40% of the *VHL* mutations are frameshift or nonsense mutations, and the remaining 40% are missense mutations [116,117].

*VHL* consists of three exons and is located in a region of approximately 13,000 bp on 3p25.3 in the human genome. However, translation is initiated by two methionine residues at amino acids 1 and 54; therefore, VHL proteins of 213 and 160 amino acids (approximately 30 and 19 kDa in size, respectively) are produced, both of which have tumor-suppressing functions [115].

VHLs shuttle between the nucleus and cytoplasm, and, at steady state, the majority of the protein resides in the cytoplasm [118,119,120,121,122,123,124,125,126]. Some VHL proteins are also found in the mitochondria and are associated with the endoplasmic reticulum [127,128]. The primary sequence of VHL contains two functional subdomains, called the alpha and beta domains (Figure 2) [129]. The alpha domain, corresponding to residues 155–192, consists of three alpha helices and forms the core of a protein complex containing Elongin C, Elongin B, Cul2, and Rbx1 (also known as Roc1) [130,131,132,133,134]. This complex possesses ubiquitin ligase activity, which leads to proteasomal degradation of the target protein [135,136].

The most well-assessed function of VHL is its function as an E3 ubiquitin ligase complex that regulates the degradation of the transcription factor HIF. The VHL protein consists of two structural functional domains, α and β. The α-domain binds to Elongin C, Elongin B, Cullin2(Cullin2), and Rbx1 to form an E3 ubiquitin ligase complex (VBC-Cul2/E3 complex) [127,133,137]. VHL binds to the target protein at the β-domain; however, one of the ubiquitinated target proteins is HIF-1α, which undergoes post-translational modification (hydroxylation of proline residues). HIF forms a heterocomplex of two molecules: HIF-1α and HIF-1β. HIF-1α binds to the cofactor CBP/p300 and has functional activity as a transcription factor. Additionally, HIF-1α is post-translationally modified by HIF prolyl hydroxylase under normoxic conditions by hydroxylating proline residues (amino acids 402 and 564 in HIF-1α, and 405 and 531 in HIF-2α). Hydroxylated HIF-α (post-translationally modified) is polyubiquitinated by the VBC-Cul2/E3 complex and then rapidly degraded by the 26S proteasome [138,139]. Conversely, under hypoxic conditions, the ubiquitination and degradation of HIF-1α are suppressed; HIF-1α translocates to the nucleus, binds to HIF-1β, binds to the hypoxia response element in the gene promoter, and initiates the transcription of various genes. [140] (Figure 4). 

### 4.2. Expression of the VHL Gene and Protein in the Nervous System

With regard to *VHL* expression, strong expression of *VHL* mRNA during human embryogenesis was found in all three germ layers; in particular, its expression was observed in the central nervous system, kidney, testis, and lung via in situ hybridization analysis at 4, 6, and 10 weeks post conception. Moreover, studies of the expression distribution pattern of VHL have shown that VHL is widely expressed in normal human tissues and that high levels of VHL are found in neural tissue. In particular, VHL expression was observed in Purkinje cells and Golgi type II cells; additionally, regional expression was observed in the cerebellum, dentate nucleus, pontine nucleus, inferior olivary nucleus of the medulla oblongata, sympathetic ganglion, mesomysium, and colonic submucosal plexus. In target organs of VHL disease, other than the nervous system, high VHL expression was observed in the renal tubular system, pancreatic exocrine gland, adrenal cortex, and liver parenchyma [141]. In addition, the expression of the *VHL* was observed to be limited to the adult and fetal brain, kidney, and adult prostate. Nerve cells, including cerebellar Purkinje cells, in adult and fetal brains were positive for *VHL* gene expression [142]. In contrast to prior studies, VHL was observed to be widely expressed in normal human tissues. Cellular distribution of the protein was confined to the cytoplasm of specific cell types. High levels of the protein were observed in neural tissue, especially in Purkinje cells and Golgi type II cells, the dentate nucleus of the cerebellum and pontine nuclei, inferior olivary nucleus of the medulla oblongata, orthosympathetic ganglia, and myenteric and submucous plexus of the colon [122,143]. In addition, strong expression of VHL was found in neural progenitor cells that differentiated into neurons on day 14 of culture [144] (Figure 5).

### 4.3. Role of VHL Protein in the Molecular Mechanisms of Neuronal Differentiation and Nervous System Development

We showed that VHL expression in rat forebrain-derived NSCs is predominant in the cytoplasm of cells expressing the neuronal marker microtubule-associated proteins 2 (MAP2) and that the forced expression of VHL rapidly induces differentiation into neurons [144]. In addition, the electrophysiological analysis revealed that *VHL*-introduced NSCs had an average maximum current of ≥4000 pA; additionally, the voltage-dependent sodium current of these cells was determined to be >4000 pA. These cells showed more than three times the average maximum current of cells that had naturally differentiated into cells and exhibited properties of electro-physiologically mature neurons, suggesting the possibility of their roles as functional neurons in vitro [145]. Furthermore, more than 50% of VHL transgenic NSCs transplanted into Parkinson’s disease rat models not only differentiated into tyrosine hydroxylase (TH)-positive dopamine-producing cells but also showed a marked reduction in apomorphine-induced turnover in behavioral analysis. Approximately 30% of the model rats showed no induced rotation [146]. This finding indicated that *VHL*-introduced NSCs function as dopamine-producing neurons in the transplanted rat brain. Therefore, neural transplantation therapy for neurological diseases, such as Parkinson’s disease, using NSCs as donors is of great importance, with VHL expression potentially contributing to the understanding of these treatments.

The VHL forms a complex (VBC-Cul2/E3 complex) with Elongin B and C, Cullin2, and Rbx1, which acts as an E3 ligase. After the ubiquitination of JAK2, its degradation by proteasome 26S, followed by the suppression of downstream STAT expression, may affect bHLH-type transcription factors, in turn affecting the Notch signaling and STAT/Smad pathways [13,14,147,148]. VBC-Cul2/E3 complexes ubiquitinate HIF-1α for proteasomal 26S degradation, followed by the activation of downstream factors of HIF-1α. The suppression of HIF-1α expression is thought to induce neuronal rather than glial differentiation [149]. The VHL forms a VBC-Cul2 complex with its binding proteins, Elongin B/C and Cul2, and degrades HIF. However, normal oxygen tension is required for the normal functioning of VHL. Under hypoxia, VHL does not function normally, and therefore, HIF is not degraded; then, hypoxia-induced VEGF, EPO, and other factors are induced [150] (Figure 4). Therefore, in our study, we placed NSCs under hypoxic conditions and investigated their differentiation when VHL did not function normally. Consequently, the percentage of the neuronal marker MAP-positive cells decreased under hypoxic conditions, but the percentage of the astroglial marker GFAP-positive cells increased. VHL protein dysfunction due to hypoxia may result in the inhibition of neuronal differentiation [151]. Since VHL does not function under hypoxia, HIF is not degraded, and downstream transcription factors are induced, suggesting that HIF may be involved in the differentiation of NSCs into glia. As parenchymal tumors of the central nervous system, such as gliomas, can arise from NSCs, this VHL-HIF transcriptional mechanism is crucial in the development of parenchymal tumors of the central nervous system. [152,153]

Furthermore, we identified that the domain of VHL involved in neuronal differentiation is the BC-box motif. Interestingly, it was found that neural differentiation occurs when the BC-box motif in the VHL protein is introduced into NSCs and other somatic stem cells (skin-derived stem cells, bone marrow-derived stem cells, adipose-derived stem cells, and human epidermal stem cells) [145,154,155,156,157,158]. In addition, in the SOCS family proteins that have a BC-box motif structure homologous to the VHL protein, most of the BC-box motifs exhibited neuronal differentiation activity [148].

It has also recently been reported that VHL interacts with Daam2; specifically, their mutual antagonism regulates differentiation in oligodendrocyte development. Proteomic analysis of the Daam2–VHL complex using a conditional gene knockout mouse model revealed that the Nedd4 complex ubiquitinates amino acid K63 and stabilizes VHL; this process is required for myelination. Furthermore, studies in mouse demyelination models and white matter lesions in patients with multiple sclerosis have demonstrated the role of the Nedd4-VHL pathway in remyelination; in particular, Nedd4 is required for differentiation and myelination of oligodendrocytes. In addition, it has been established that VHL-HIF signaling plays a key role in the tumorigenesis of a wide range of malignancies. The Daam2–VHL relationship was first discovered in gliomas, with Nedd4 being a key upstream regulator of VHL. This discovery may have broad implications, including both CNS and non-CNS malignancies. It is important to clarify whether Nedd4-VHL-HIF signaling is involved in the development of renal carcinoma and hemangioblastoma, which commonly occur in VHL disease [159].

Recently, attention has been focused on the role of HIF-1α in NSCs during nerve regeneration after traumatic brain injury and stroke; furthermore, VHL, upstream of HIF-1α, is thought to play an important role in this process. Endogenous NSCs and neural progenitor cells (NSPCs) have been shown to promote repair after brain injury and stroke. Therefore, these cells are therapeutic targets for promoting cell replacement therapy and neuroplasticity. The mechanisms supporting NSPC survival and recruitment of damaged cells in the ischemic brain have highlighted a novel role for HIF-1α in NSPC function. HIF-1α mediates adaptive cellular responses to hypoxia through the direct transcriptional regulation of cell metabolism and angiogenesis; however, recent studies have suggested that it may be mediated by regulation of the Notch and Wnt/β-catenin signaling pathways [152].

A novel role of HIF-1α in stem cell differentiation has also been demonstrated. HIF-1α functions as a key mediator of NSPC function under normal conditions and in stroke, playing a central role in regulating NSPC responses to hypoxia, metabolism, and the maintenance of the vascular environment of the NSC niche. The role of VHL proteins upstream of HIF-1α has also attracted attention. Focal cerebral ischemia stimulates the proliferation and migration of SVZ-derived progenitor cells into the lesion, suggesting that NSPCs in the SVZ recruit new oligodendrocyte progenitor cells, astrocytes, and neural progenitor cells to the peri-infarct region. In addition, HIF-1α not only maintains the vascular niche environment and promotes angiogenesis through transcriptional regulation of VEGF but also is involved in determining the pluripotency and differentiation of NSPCs. As HIF-1α plays an important role in such processes, VHL, upstream of HIF-1α, is thought to play a similarly important role [160].

## 5. Conclusions

bHLH factors play a central role in mammalian nervous system development and differentiation from NSCs to nervous system cells. The Notch and STAT/Smad pathways are thought to be involved in this process; additionally, the STAT transduction pathway is regulated by SOCS and VHL proteins. NSCs differentiate into three tumor cell lineages: neurons, astrocytes, and oligodendrocytes. SOCS is involved in neuronal differentiation, and VHL is involved in differentiation into neurons and oligodendrocytes; however, both proteins function to suppress astrocyte differentiation. SOCS and VHL proteins contain homologous structures comprising a BC-box motif, called SOCS box and VHL box, respectively. SOCS proteins specifically bind to Elongin B/C-Cul5-Rbx2 to form an SBC-Cul5/E3 complex, whereas VHL specifically binds to Elongin B/C-Cul2-Rbx1 to form a VBC-Cul2/E3 complex. Both complexes act as E3 ligases that target proteins via the ubiquitin–proteasome system, ultimately resulting in the degradation and inhibition of downstream transduction pathways. We have shown that SBS-Cul5/E3 and VBC-Cul2/E3 recruit JAK and HIF as their primary target proteins, respectively; however, JAK is also a target protein of VBC-Cul2. Additionally, SOCSs not only act on the ubiquitin–proteasome system but also act directly on JAKs to suppress the JAK-STAT pathway. SOCS and VHL proteins are expressed in the nervous system, with predominant expression in embryonic brain neurons. Both SOCS and VHL proteins promote neurite outgrowth and induce neuronal differentiation. SOCS proteins are involved in differentiation into neurons, and VHL proteins are involved in differentiation into neurons and oligodendrocytes. In addition, SOCS and VHL proteins function as tumor suppressors; consequently, it has been suggested that the inactivation of these proteins may lead to the development of nervous system malignancies. The mechanism of action of SOCS and VHL proteins involved in neuronal differentiation and development of the nervous system is thought to be mediated via the inhibition of downstream signaling pathways (JAK-STAT and HIF-VEGF pathways). In addition, because SOCS and VHL proteins promote nerve regeneration, they are expected to be applied in nerve-regenerative medicine for traumatic brain injury and stroke.

## Figures and Tables

**Figure 1 ijms-24-03880-f001:**
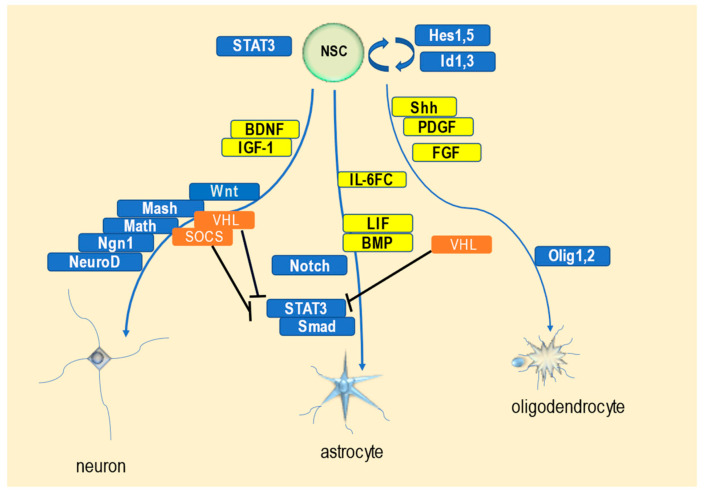
Molecular differentiation pathways of neural stem cells (NSCs). NSCs are maintained by HES1 and 5, ID1 and 3, and the signal transduction and activator of transcription 3 (STAT3). Neuronal differentiation is induced by extracellular factors, such as platelet-derived growth factor (PDGF), brain-derived neurotrophic factor (BDNF), and insulin-like growth factor-1 (IGF-1), and intracellular factors, such as Wnt, Mash, Math, neurogenin1 (Ngn1), von Hippel-Lindau (VHL), and suppressor of cytokine signaling (SOCS). Astrocytic differentiation is induced by extracellular factors, such as interleukin-6 family cytokines (IL-6FCs), leukemia inhibitory factor (LIF), and bone morphogenetic protein (BMP), and intracellular factors, such as Notch, STAT3, and small mother against decapentaplegic (Smad). Oligodendrocytic differentiation is induced by extracellular factors, such as Sonic Hedgehog (Shh), PDGF, fibroblast growth factor (FGF), and bone morphogenetic protein (BMP), and intracellular factors, such as VHL and Olig1 and 2. SOCS and VHL inhibit STAT3 through the Janus kinase–signal transduction and activator of transcription (JAK-STAT) pathway [2,3,4,5,6,7,8,9,10,11,12].

**Figure 2 ijms-24-03880-f002:**
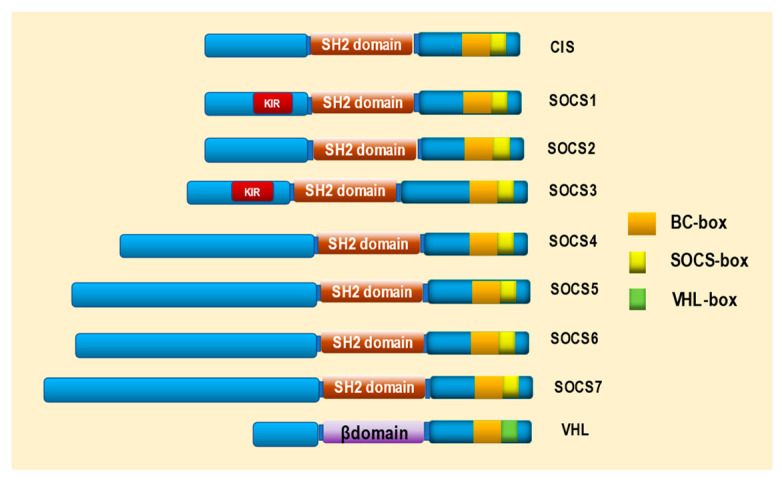
Structure of suppressor of cytokine signaling (SOCS) and von Hippel-Lindau (VHL) proteins. SOCS family proteins, cytokine-inducible Src homology 2 (SH2)-containing protein (CIS), and SOCS1–7 contain an SH2 domain, BC box, and SOCS box. SOCS1 and SOCS3 also contain a kinase inhibitory region (KIR). VHL contains a β domain, BC box, and VHL box.

**Figure 3 ijms-24-03880-f003:**
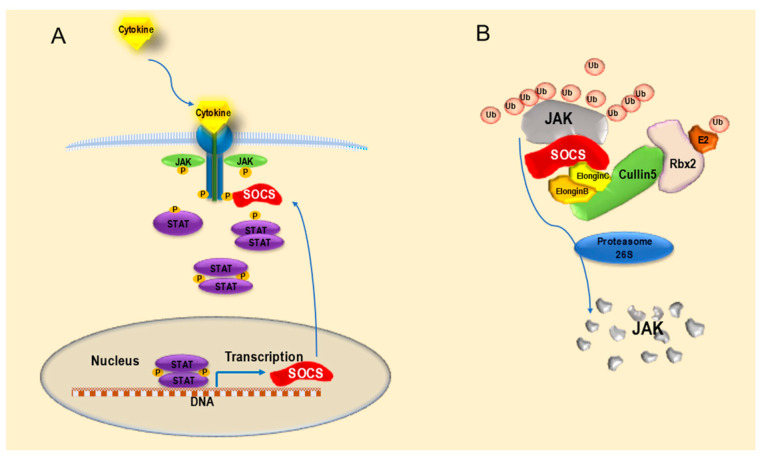
Molecular mechanism of the suppressor of cytokine signaling (SOCS) proteins. (**A**) After cytokines bind to its receptor, the Janus kinase–signal transduction and activator of transcription (JAK-STAT) pathway is activated, and the signal transduction and activator of transcription (STAT)-mediated expression of SOCS is induced. In turn, SOCS inhibits the JAK-STAT pathway in a negative feedback loop. (**B**) SOCS protein recruits Elongin C, Elongin B, Cullin5, Rbx2, and E2 and forms a SOCS/E3 complex. Thereafter, the SOCS/E3 complex binds Janus kinase (JAK); JAK is then ubiquitinated and finally degraded by proteasome 26S.

**Figure 4 ijms-24-03880-f004:**
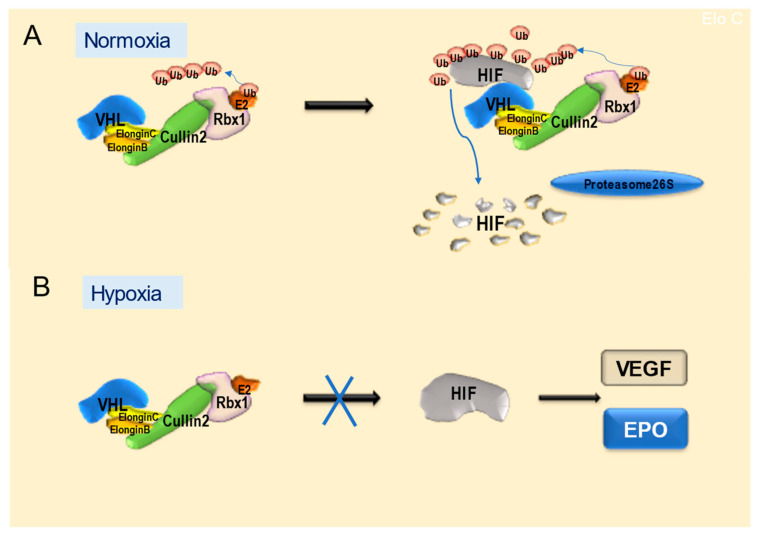
Molecular mechanism of von Hippel-Lindau (VHL) protein. (**A**) Under normoxia, VHL recruits Elongin C, Elongin B, Cullin2, Rbx1, and E2, forming a VHL/E3 complex. Then, VHL binds hypoxia-inducible factor/Janus kinase (HIF/JAK), resulting in HIF/JAK ubiquitination and degradation by proteasome 26S. (**B**) Under hypoxia, the ubiquitination and degradation of HIF-1α are suppressed; HIF-1α translocates to the nucleus, binds to HIF-1β, binds to the hypoxia response element in the gene promoter, and initiates the transcription of various genes, such as *VEGF* and *EPO* [138,139,140].

**Figure 5 ijms-24-03880-f005:**
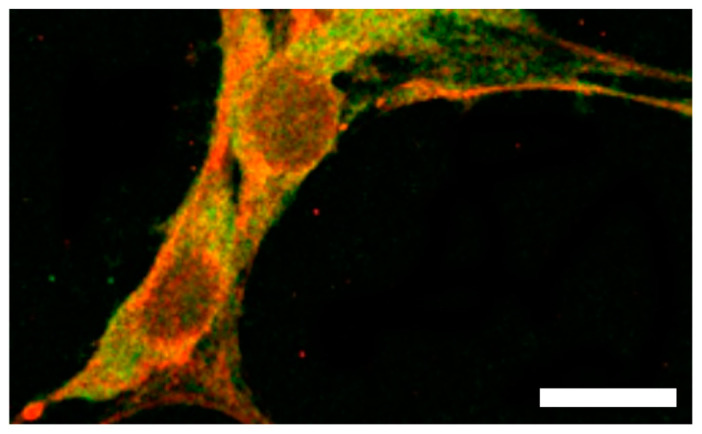
Double-fluorescence immunocytochemical image of neuronal progenitor cells on day 14 of culture, obtained using a laser scanning confocal microscope. Von Hippel-Lindau (VHL) protein and microtubule-associated proteins (MAPs) are co-expressed in neuronal progenitor cells. VHL is detected with fluorescein isothiocyanate (FITC, green), and MAPs are detected with rhodamine (red). Scale bar = 10 μm.

## Data Availability

The data presented in this study are openly available only in this text.

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
