# Peer review of "Role of SOCS and VHL Proteins in Neuronal Differentiation and Development"

_ijms, 2023, doi:10.3390/ijms24043880_

Round 1

Reviewer 1 Report

The review submitted by Hiroshi Kanno and co-authors entitled " Role of SOCS and VHL proteins in neuronal differentiation and development", focus on the role of both SOCS and VHL proteins in neuronal differentiation and neurite outgrowth. This work should be of wide interests to most researchers on neuroscience and molecular medicine etc. The review is well structured and annotated with several relevant figures to annotate the key issues of the review. This manuscript is well written, organized and sounds with a good standard of English language. The following points need to be addressed:

All Figures have to be referred within the main text.

Figure 1: the role of Shh, PDGF and FGF is not mentioned in the figure legend. Also the authors must include the corresponding references in figure legends.

Figure 2 is not referred in the main text. In addition is missing a sentence in the main text with how many mammalian members belongs to the SOCS family. 

Figure 5 is missing the Scale bar from the image and the figure legend.

The authors should include one last image with the possible mechanisms used by SOCS and VHL proteins implicated in neuronal differentiation, focusing on the upstream and downstream target genes refereed to the main text. This will help the readers to understand the review easily since it is too wordy in terms to understand and take the key points in its present format.

Typing error:

Line 106-107: “HIF-2α, whereas Cul5–Rbx2 knockdown did not affect the function of the Cul2–Rbx1 and Cul5–Rbx2 modules.[13].” Please delete the dot before the reference.

Line 160-162: “Conversely, SOCS3 inhibits the The nuclear factor κB (NF-κB) signaling pathway, antagonizes cyclic AMP (cAMP)-mediated signaling, and promotes signaling via the microtubule affinity-regulating kinase (MAPK) pathway [80].” Please delete “The”.

Author Response

The author's reply to review report is descibed in the attached file.

Reviewer 2 Report

This review is very interesting review and provides useful information about the role of SOC and VHL in neuronal differentiation and development. The manuscript is written well and acceptable for publication but need minor revision.

1)     Manuscript language can be improved.

2)     The manuscript needs minor correction, spelling, spacing etc.

3)     On page 10, line 438, the author mentioned that NCS differentiate into three tumor cell linage: neurons, astrocytes, and oligodendrocytes. Is this tumor or neuronal cell lineage?

4)     Schematic representation needs some improvement.

5)     Endnote should be used for citation and reference if not used.

Author Response

The author's reply to the review report (Reviewer 2) is described in the attached file.
